# Clinical Field and Alternative Clinical Practice Experience in a Pandemic Situation of Nursing Students Who Have Experienced Clinical Practice before COVID-19

**DOI:** 10.3390/ijerph192013372

**Published:** 2022-10-16

**Authors:** Hyeran An, Sunnam Park, Jongeun Lee

**Affiliations:** 1Department of Nursing, College of Nursing, Daegu Catholic University, Daegu 42472, Korea; 2Department of Nursing, Seoul Women’s College of Nursing, Seoul 03617, Korea; 3Department of Nursing Science, College of Medicine, Chungbuk National University, Cheongju-si 28644, Korea

**Keywords:** clinical practicum, nursing students, COVID-19, pandemic, nursing education

## Abstract

This study aimed to understand the experiences of nursing students who experienced clinical practice before the outbreak of COVID-19 on clinical field practice and alternative clinical practice adopted during the COVID-19 pandemic. A phenomenological study was conducted on 14 graduates who experienced clinical field practice and alternative clinical practice during a pandemic. Data were collected using individual in-depth interviews that were semi-structured. Data were analyzed according to Colaizzi’s procedure. As a result of data analysis, five themes and 17 sub-themes were selected. The five themes were: alienation during the process of clinical practice change, regret caused by alternative clinical practice, alternative clinical practice as a supplementary measure, difficulties due to COVID-19, non-replaceable clinical field practice. It is necessary to consider using alternative clinical practices to complement the clinical setting and improve the quality of clinical practice in the post-corona era. To this end, it is necessary to supplement the disadvantages of alternative clinical practice, such as reduced concentration and lack of sense of presence, by applying integrated education using Edutech.

## 1. Introduction

The Korean government implemented the With-Corona policy in November 2021. Therefore, social distancing was lifted on 18 April 2022, after about two years and one month since the pandemic [1]. Subsequently, the gradual recovery of daily life from the coronavirus disease 2019 (COVID-19) began. Clinical practice was suspended or limited worldwide in 2020 due to the COVID-19 pandemic [2]. However, the proportion of clinical practice replaced by alternative clinical practice has increased as the hospital setting is stabilized through systematic quarantine management. A clinical setting is a learning place for future nurses to establish their professional identities as well as a workplace where they will take care of patients as nurses in the near future. In addition, nursing students gain an understanding of the essence of nursing through clinical experience using their theoretical knowledge [3]. Therefore, clinical practice is essential for nursing students who need to have the ability to care for a variety of patients [4]. However, problems existed in the clinical setting before the COVID-19 outbreak, such as unstable clinical practice conditions, limitations in the roles of student nurses, and excessive burden [5] due to the quantitative expansion of the nursing department’s quota and the improvement of consumer rights in the clinical setting [6], which remain unresolved.

In the transitional period occurring just before the COVID-19 outbreak, when simulation-based learning that could safely implement situations similar to clinical settings was gradually expanded as an alternative [7], COVID-19 played a major role in helping to change the existing school class management style [8]. In 2020, as the Korean accreditation board of nursing education allowed the replacement of 1000 h of clinical practice per student with alternative clinical practice [9], most nursing colleges adopted alternative clinical practice in various ways, such as watching videos, watching videos on core basic nursing skills, online simulations, and virtual simulations using web-based simulation platforms [4]. If the change in the educational paradigm began due to the fourth industrial revolution, digital transformation was accelerated due to COVID-19. In other words, it was an opportunity to promote the transition to a new education paradigm away from the existing education style [8]. Recently, the possibility of using edutech is gradually increasing in education for clinical site practice [8]. Edutech is a compound word for education and technology and refers to a new educational paradigm that combines information and communication technology with education, such as big data, artificial intelligence, robots, and virtual reality [10].

In order to make changes in clinical practice caused by COVID-19 and have an opportunity to improve the quality of clinical practice education, it is necessary to prepare a clear direction and a specific roadmap for clinical practice education [11]. In addition, when many experts are warning of a future pandemic caused by other infectious diseases [12], it is necessary to establish a stable clinical practice system so that the same problems are not repeated. To this end, a systematic discussion about the problems, limitations, and effects of clinical practice and alternative clinical practices should be had before returning clinical practices to pre-COVID-19 pandemic clinical practices. The phenomenological method [13] to identify and analyze the research phenomenon is suitable for minimizing researchers’ prejudice and confirming the essence of clinical field practice and alternative clinical practices in a pandemic.

Publications on domestic and foreign qualitative research methods related to the clinical practice of nursing students in the current COVID-19 situation were published on clinical field practice experience [13,14,15,16], online clinical practice experience [4,17,18], virtual reality-based simulation clinical practice experience in one subject [6], online clinical practice and in-school clinical practice experience [19]. Clinical practice experience in clinical field practice or alternative clinical practices was limited in each partial aspect. Therefore, it is necessary to do an in-depth assessment of the clinical field practice experiences and alternative clinical experiences of nursing students, who are the main body in the clinical setting, from a holistic perspective. To this end, this study intends to provide useful basic data for strategies to improve the quality of clinical practice during the COVID-19 era and beyond by applying a phenomenological method to identify the essence of clinical field practice and alternative clinical practices in a pandemic situation experienced by nursing students before and after COVID-19.

This study aimed to gain a comprehensive and in-depth understanding of nursing students’ experiences in the clinical setting with regard to clinical field practice and alternative clinical practices before and during the outbreak of COVID-19 through the phenomenological method.

## 2. Materials and Methods

The authors followed the Consolidated Criteria for Reporting Qualitative Research (COREQ) reporting guidelines [20].

### 2.1. Design

This study is a phenomenological study to explore and describe the meaning of nursing students’ experiences in clinical field practice and alternative clinical practices during a pandemic situation caused by the COVID-19 pandemic. The phenomenological method is an appropriate research method to identify and describe the participant’s experience from the participant’s point of view and to clarify the nature and structure of the experience [21].

### 2.2. Participant Recruitment

Nursing graduates who had clinical experience before and after the COVID-19 outbreak, as juniors (2019) and seniors (2020) at the nursing schools, respectively, were recruited in this study. The Korean nursing education academic system was unified into a four-year system with the revision of the Higher Education Act in 2011, laying the legal basis for standardizing nursing education [22]. Nursing students take theory classes in their first year, theory classes and in-school practice in their second year, and theory classes, in-school and clinical practice in their third and fourth years. In this study, nursing graduates who had no clinical field practice experience before the COVID-19 outbreak or no clinical alternative practice experience after the COVID-19 outbreak were excluded from the participants. The purpose and methodology of this study were explained to the professors of each of the three participating nursing schools (C, I, and S). These professors recommended nursing graduates who were willing to describe their first-hand clinical experiences during the COVID-19 pandemic. The recommended graduates (February 2021) were directly informed of the objectives and methods of the study by the researcher, and a total of 14 graduates (four or five graduates per school) who agreed to participate in the study were finally included.

### 2.3. Data Collection

Data were conducted between August and October 2021 via individual in-depth interviews. Information-rich cases were selected using purposive sampling until the point of data saturation [23]. One-on-one in-depth interviews took place either offline in the researcher’s laboratory, in order for the interviewee to talk comfortably in a quiet environment, or online via Zoom, a video conferencing platform, for participants who preferred a contactless interview due to COVID-19. According to the choice of participants, 3 out of 14 participants had face-to-face interviews, and 11 participants had interviews online via Zoom. The face-to-face interviews were conducted at a time when it was compulsory to wear a mask, so there was a limitation in confirming the participant’s facial expressions, but it was easy to confirm non-verbal messages such as gestures and attitudes. In the case of online interviews through Zoom, it was easy to check their facial expressions because they did not wear a mask, but the flow of the interview was sometimes interrupted if the participants did not have a smooth internet connection. Each session lasted 50–80 min, and each participant was interviewed once or twice. A semi-structured interview format was used for data collection, and the questions of the initial interview are included in Table 1. During an interview, the interviewee’s non-verbal reactions, such as facial expression, gaze, body gestures, and voice tremors, were recorded as field notes. All the interviews were recorded with consent. The recordings were transcribed verbatim by a research assistant and checked by the researcher. A second one-on-one interview was arranged with three participants via Zoom to clarify unclear statements from the first interview or check additional facts necessary for the analysis.

### 2.4. Data Analysis

Data were analyzed in a circular fashion to understand and describe the participants’ essential clinical practice experiences using the phenomenological method developed by Colaizzi [21]. The analysis method of Colaizzi [21] is considered to be the most suitable for stating essential experiences by deriving common themes that appear in nursing students’ experiences in the clinical field and alternative clinical practice and integrating them. The data analysis included the following seven procedural steps: (1) We read the transcript of each in-depth interview multiple times to grasp the overall atmosphere of each interviewee’s experience, (2) while repeatedly reading the transcript, the researcher extracted significant or repetitive statements as major statements reflecting the essential aspects of clinical experience amidst the COVID-19 pandemic, (3) formulated meanings of the interviewee’s experience were derived from the major statements by discerning their abstract meanings, (4) statements with similar meanings were categorized into 17 sub-themes, which were then re-categorized into five themes, (5) an exhaustive description was drawn from the themes and interviewees’ statements that appropriately reflected these themes, (6) the exhaustive description was thoroughly reviewed to understand the essence of the interviewee’s experience, and (7) for reliability and validity testing of the analysis results, a co-researcher (experienced in phenomenological qualitative analysis) and three participants also reviewed the results.

### 2.5. Rigor

The following were enhanced to ensure rigor, credibility, transferability, dependability, and confirmability, which are the four major evaluation criteria for rigor in qualitative research [24]. First, to enhance credibility, participants with experiences that reflect well the research phenomenon were selected, and the interview transcripts were thoroughly checked by the researcher to minimize false and missing data. Second, to enhance transferability, participants were recruited via purposive sampling. Data collection continued until theoretical saturation was reached and the general characteristics of participants were presented. Additionally, the data analysis results were examined by three of the participants to check their accuracy and consistency with the participants’ actual experiences. Third, to enhance dependability, data analysis was performed in strict compliance with the method proposed by Colaizzi [21], and the entire study process and results were evaluated by a nursing professor experienced in qualitative research to verify the consistency of the study. Fourth, to enhance confirmability, much effort was put into a faithful and vivid description of the participants’ experiences as phenomenological inquiry and precluding researcher bias by recording the researcher’s preconceptions and assumptions in the journal throughout the study process.

## 3. Results

As a result of the phenomenological analysis of nursing students’ experience in clinical practice and alternative clinical practice, 17 sub-themes and 5 themes were extracted (Table 2). Table 3 shows the general characteristics of participants in this study.

### 3.1. Alienation during the Process of Clinical Practice Change

#### 3.1.1. Frustration Aggravated by Lack of Communication

Most participants said they were angry and frustrated with the university that asked them to recklessly wait, even though they understood the university’s confusion during the period when the start of the semester was delayed due to the unprecedented COVID-19 situation. Even in the abrupt changes of clinical practice plans due to COVID-19, students felt helpless as they were informed of the university’s decision that only considered the situation at the clinical sites rather than the students’ opinions.


*“We are not a five-minute stand-by option of the school. At the very least, they should have come up with a statement like ‘We were considering a way out of this crisis among these alternatives’ or just ‘What are your opinions?’ Yes, just asking us in that process would have made us less frustrated and angry.”*
(Participant 11)

#### 3.1.2. Clinical Practice Took Priority over Student Safety

Participants wondered whether it was right for their schools to take the risk of sending students to clinical practice without realistic and systematic coping strategies, vaccination, and/or treatments. Some of the participants also mentioned that the most important thing in such a situation was the safety of the patients and students, but the school prioritized clinical practice over student safety.


*“Hospitals are the most vulnerable places to contract COVID-19 because of patients, and the possibility of infection in clinical practice seemed too risky. We thought we knew better, but the school seemed to ignore the situation.”*
(Participant 4)

### 3.2. Regret Caused by Alternative Clinical Practice

#### 3.2.1. Repetition Rather Than Critical Thinking in Virtual Simulation

The participants said that the virtual simulation program they encountered as an alternative clinical practice was initially interesting. However, because it was virtual rather than real and repeated learning was possible, the participants gradually depended on the program’s feedback to find the correct answer. Therefore, the learning effect was decreased.


*“I must have thought about it and hesitated because I was afraid that it would go wrong if I did it to an actual patient… but I’m not doing it for actual patients, so I tried it once, and if I’m wrong, I saw the feedback and tried again, and if I repeated that, I could reach 100%.”*
(Participant 7)

#### 3.2.2. Monotony Due to Fixed Answers in Case Study

For most participants, a case study in clinical practice was the process of collecting information directly from the patient and caregiver and finding the optimal answer derived by the students themselves. On the other hand, the alternative clinical practice was a learning process with a clear limitation that the correct answer was set by the instructor. Participants showed regret for the lost opportunity of learning diverse cases because the case studies in the alternative clinical practice had to be performed only on predetermined subjects.


*“In clinical practice, I could also learn about various other case study subjects chosen by other students. I missed this opportunity in the alternative clinical practice because everybody wrote on the same subject.”*
(Participant 5)

#### 3.2.3. Difficulty in Concentration during Watching the Learning Video

Participants said they did not watch the learning videos but found content related to the video on the internet and wrote and submitted reports as if they had watched the video. In addition, participants found that the video materials used in the alternative clinical practice were too antiquated to properly reflect reality or lacked relevance to the learning theme and thus were not helpful enough to achieve the desired learning outcomes.


*“Usually, I watched the videos at 2x speed or did not even watch them at all. I looked up the contents on the internet, read them and wrote them down. And then I realized that it was a waste of time.”*
(Participant 2)


*“Some videos were made too long ago, while others lacked relevance to the theme. Unfortunately, none of them were suitable for learning.”*
(Participant 5)

#### 3.2.4. Regret the Lost Opportunity to Gain Various Experiences

Most participants were limited to specific departments such as the operating room (OR), emergency room (ER), and intensive care unit (ICU) and proceeded with alternative clinical practice. Therefore, even after completing the alternative clinical practices, participants expressed concern that they knew very little about the specific departments. In addition, the participants felt that it was regrettable that the opportunity to plan for the future as a nurse had disappeared by directly experiencing various wards and comparing which wards are suitable for them.


*“It was an opportunity for me to try out various wards and find out which ward suits me best, but it was so sad that the opportunity seemed to have been*
*lost.”*
(Participant 8)


*“I couldn’t go to the OR, so I did the clinical practice online, but even after the clinical practice was over, I didn’t even know what kind of place it is.”*
(Participant 10)

### 3.3. Alternative Clinical Practice as a Supplementary Measure

#### 3.3.1. Served as a Preparation Step before Clinical Field Practice

The participants expressed that the lack of clinical practice in the clinical setting was supplemented by alternative clinical practice. Conversely, the lack of alternative clinical practice could be substituted, with clinical practice courses with one week of practice in the clinical setting and one week of on-campus training (an alternative clinical practice), instead of two weeks of clinical practice in the clinical setting due to COVID-19.


*“Clinical practice is a chance event. Sometimes we learn many different cases, other times not at all. In this sense, it was better to do both clinical practice and on-campus clinical practice for one week each because I could make up for the deficit in clinical practice with on-campus clinical practice.”*
(Participant 2)

#### 3.3.2. Feeling Interested as the Main Body of Clinical Practice

Alternative clinical practices were provided in various formats such as offline or online simulations of nursing scenarios (vSim for Nursing), PBL, CBL, video material, case studies, etc. Participants expressed their satisfaction with simulations and other learning platforms where they could perform nursing interventions themselves instead of visual observation only.


*“In general, the alternative clinical practice plan was densely scheduled with practical sessions and used diverse learning tools, including offline simulation and vSim simulation, video materials, report writing, and so forth. In a way, it was better than idly standing in the hospital.”*
(Participant 5)


*“In clinical field practice, I was just an observer, watching what the nurse mentor was doing, but the simulation was fun because I could directly practice my nursing skills on the patient.”*
(Participant 10)

#### 3.3.3. Satisfaction with a Fair Evaluation

The participants had doubts about the fairness of the evaluation made by the head nurse (the clinical practice leader) because they mainly learned by following the nurses who took care of the patients directly. They expressed that even if they pursued the nurses diligently and responded kindly to the patients and caregivers, they would be immediately evaluated and receive an incomprehensible evaluation if they encountered the head nurse during a brief break. On the other hand, they were satisfied that they were evaluated objectively because the professor assessed all students equally and gave feedback on the alternative clinical practice results.


*“There were many times when I could not understand*
*why I*
*received a certain score*
*during clinical practice. However, the professor directly evaluated and gave feedback, so I felt I was evaluated fairly*
*for the alternative clinical*
*practices.”*
(Participant 8)

#### 3.3.4. Less Burden Due to Shortened Clinical Field Practice Period

Most participants mentioned that during clinical field practice, there was more pressure. Participants were constantly being evaluated by nurses and the patients, and the fatigue and stress levels increased due to unreasonable demands from the caregivers (families or paid hospital caregivers). They also mentioned that the burden further increased due to the fear of negative evaluation results that may be disadvantageous to their schools, fellow students, and alumni.


*“Whenever I was in the ward, I felt like I was being observed because there were always nurses and caregivers. I was feeling tense, thinking that I always have to do my best to make sure I would not harm my school reputation and my fellow students and alumni and that I have to make a good impression on them.”*
(Participant 1)

### 3.4. Difficulties Due to COVID-19

#### 3.4.1. Anxiety about Becoming a Spreader

Most participants showed a vague belief that they would be safe because they were young and healthy and did not feel threatened by COVID-19 infection. Instead, the participants were worried about becoming a disease carrier and unknowingly spreading it to others, such as patients with an impaired immune system, caregivers, nurses, and friends.


*“Being young and healthy, I thought I would be safe. I was more worried about unknowingly passing it on to others as an asymptomatic carrier than about getting infected myself, and I still am...”*
(Participant 1)

#### 3.4.2. Blurred Awareness in the Clinical Setting

Some participants mentioned that their risk awareness gradually decreased during clinical practice due to long working hours and having to always be on alert for the needs of nurses and patients. This led to occasional non-compliance with infection prevention measures, particularly when they were with friends.


*“I became increasingly lax once my clinical practice training began, easily dismissing the thought of infection for myself and my friends. So, I became less and less cautious when I was with my friends; for example, strapping my mask across the chin.”*
(Participant 10)

#### 3.4.3. Inconvenience Caused by Wearing Protective Equipment

Due to prolonged usage of KF94 masks during clinical field practice, participants complained of discomforts such as difficulty in breathing, pain, and blurred vision (because of foggy goggles). Additionally, most of the participants complained of difficulties in verbal and non-verbal communication with nurses and patients during clinical field practice due to the use of masks. Since half of the face was covered by a mask, the facial expressions of nurses and patients were difficult to understand. Therefore, participants had to be more attentive during communication.


*“I wear glasses, but I also wear goggles and a mask, so it was very inconvenient because my eyes suddenly became blurry, and thus, my vision was not completely secured. In addition, it hurts a lot because the back of the ear is constantly pressed…It was the most inconvenient thing during clinical practice.”*
(Participant 1)


*“Behind a mask, I could not read the facial expressions of nurses and patients, and I was constantly on alert to guess their moods or intentions.”*
(Participant 2)

#### 3.4.4. Exclusion from the Clinical Setting

Most participants revealed that they were asked to monitor the situations passively to minimize contact with patients. Hence, actual learning opportunities in the clinical setting were limited, and the free time created due to such practices was spent on doing assignments in the conference rooms.


*“On the whole, I was under the impression that there were more vacant hours in the ward or unit placement schedule compared to the pre-COVID-19 period. We were then given passive instructions such as studying with handbooks in the nurses’ station or conference room.”*
(Participant 3)


*“Earlier, I could freely enter the testing room to accompany a patient, but I had fewer opportunities to go there after the COVID-19 outbreak.”*
(Participant 4)

### 3.5. Non-Replaceable Clinical Field Practice

#### 3.5.1. Self-Awareness of Necessity

Most participants found on-site clinical practice indispensable because it provides opportunities to observe, listen, and learn up-to-date knowledge and skills that cannot be learned through handbooks and classroom lectures. Further, they re-iterated that any amount of explanation about the role of a nurse is not sufficient to know all aspects of a nurse’s role unless it is directly experienced in the clinical practice.


*“What we hear from professors isn’t always up to date. It is in the hospital that you can see how patients are taken care of in reality and how tests are performed. Many parts make me awestruck.”*
(Participant 1)


*“During my first attempt at measuring a patient’s blood pressure, I could not hear at all. I think one attempt on a patient is a lot more helpful than 100 attempts on a simulator.”*
(Participant 7)

#### 3.5.2. Learning to Build Different Relationships

Participants thought that, as future nurses, it was important to protect them from COVID-19 infection during clinical practice so that patients can be protected. In particular, they said that clinical practice is a significant opportunity to learn how to form a rapport with patients for therapeutic intervention.


*“Hospital placement is an extremely important opportunity to accumulate experience and skills necessary for communicating with patients and building rapport with them. Only such on-site experience will help us learn how to deal with patients appropriately as novice nurses without being embarrassed.”*
(Participant 10)

#### 3.5.3. Establishment of Identity as a Future nurse

Participants thought that, as future nurses, it was important to protect them from the COVID-19 infection during clinical practice so that the patients can be protected.


*“As a nurse, protecting myself is ultimately protecting others. It was through clinical practice that I came to think that protecting myself is eventually protecting others.”*
(Participant 1)

Participants also felt proud as future nurses as they perceived peoples’ positively changed attitudes toward nurses through the media coverage of their devotion and professionalism while silently working in the challenging COVID-19 pandemic setting.


*“I see a positive side of COVID-19 in that people take an interest in nurses’ professionalism and recognize their hard work and devotion, which makes me feel proud.”*
(Participant 2)

Participants also wished to emulate nurses when they observed their dedication towards the patients. Hence, clinical practice was an important learning phase where the participants could visualize their own future.


*“Seeing her paying attention to each patient, I found her really cool. I thought I would like to be like her when I become a nurse later.”*
(Participant 4)

## 4. Discussion

This study intended to explore the experiences of clinical field practice and alternative clinical practices in the pandemic situation of nursing students who experienced clinical practice before the outbreak of COVID-19.

In theme 1, “alienation during the process of clinical practice change,” participants were frustrated and anxious due to unclear communication between the university and hospital in the first semester after COVID-19. The participants’ frustration with the university was aggravated as opportunities for students to participate in discussions about responses, such as the timing and method of clinical practice, were limited [25]. In addition, they complained that they had no choice but to helplessly accept the unreasonable demands of the university and hospital by accommodating the training started without clear measures. Clinical practice, which started without clear measures when knowledge about COVID-19 was limited, except for information through the media [26], aggravated the participants’ anxiety. The university and hospital need to provide the latest information on the COVID-19 situation and countermeasures in a timely manner to alleviate students’ stress [26] and anxiety due to the uncertainty in clinical practice. In addition, it is necessary to provide opportunities for students, who are the main bodies of clinical practice, to participate in the process of discussing responses to situations from the initial stage as members of the university.

In theme 2, “regret caused by alternative clinical practice” confirmed the aspects that should be supplemented for alternative clinical practice in the post-COVID-19 era. Even though virtual simulation cannot completely replace clinical practice in a realistic and useful way in nursing practice education, it is an educational program that can be recommended as an alternative clinical practice [27]. Virtual simulation has the advantage that it can be repeated and receive immediate feedback [28]. However, this study confirmed that there is a possibility that the learning effect may be hindered by solving problems efficiently after receiving feedback without considering it carefully. In online clinical practice, the role of the instructor is important, especially the interaction between the instructor and students [4]. When the virtual simulation program is utilized, instead of relying solely on the program, the instructor should strive to enhance the learning effect by providing debriefing opportunities for students to think on their own and fully exchange their knowledge. Nursing students evaluated alternative clinical practices by comparing their experiences of alternative clinical practices versus standard clinical practice. In particular, nursing students regretted losing the opportunity to determine which department is suitable for them through clinical practice experience when selecting a hospital department to work in after graduation. Alternative clinical practice through on-campus or online clinical practice is not an actual clinical site where various situations occur. Hence, it is insufficient to feel tension or realism [8]. When developing an alternative clinical practice program, the instructor should work together with the field leader to reflect on the situation of various clinical fields. At the same time, institutional efforts such as systematic and qualitative construction of an intelligent education environment based on advanced technology [29] are necessary to compensate for the lack of realism in alternative clinical practice. Due to the fourth industry, digital transformation is taking place throughout education, and the possibility of using Edutech is gradually increasing in education for clinical site practice [8]. Due to COVID-19, the virtual world of Metaverse makes it possible for you to experience various experiences in a virtual space, away from the reality where face-to-face education is challenging [8]. Suppose the university’s financial support and instructors’ efforts to develop content are added. In that case, it is expected that Metaverse will be able to compensate for the problem of alternative clinical practice, which lacks tension and realism compared to clinical practice.

In theme 3, “alternative clinical practice as a supplementary measure”, confirmed the positive aspects of alternative clinical practice compared to clinical practice. The participants were satisfied with the escape from the chronic problems related to clinical practice that existed before COVID-19, such as unstable clinical practice conditions, heavy burden, limitations in the role of nursing students, and unguaranteed rights of clinical practice students [5], as the clinical practice conducted in the clinical setting was changed to an alternative clinical practice due to COVID-19. The participants said “the lack of learning due to the decrease in opportunities for direct clinical practice with patients in the clinical setting before COVID-19” [2] could be substituted by performing virtual and direct nursing interventions through alternative clinical practice applied with various teaching methods, such as simulation, online virtual nursing simulation, problem-based learning (PBL), case-based learning (CBL), case studies and video watching. Therefore, the positive aspect of alternative clinical practice perceived by participants stems from the problems of the existing clinical practice. Even if clinical site practice is fully resumed in the post-corona era, it is worth using alternative clinical practice as a supplement. Even though there are learning goals that students must achieve in the clinical setting, it is not possible to experience and learn everything during clinical practice [15]. A previous study [2] recommended including simulation in the clinical practice curriculum to reduce the instability of clinical practice and provide stable, practical education. Instructors should try to develop various alternative practice programs and content that can be included in clinical practice by reflecting on the times and needs of students and providing them to students.

In theme 4, “difficulties due to COVID-19”, was identified by most participants as they were concerned that they might infect others as COVID-19 spreaders rather than being concerned with their own COVID-19 exposure in the hospital environment [16]. In other words, it is considered that participants felt more anxious about social stigma as nursing students who were about to get a job as they encountered doctors and nurses [30] who worked at the frontline of the COVID-19 pandemic through media in a social atmosphere [31] to blame for COVID-19 infection on individuals or groups. In addition, the university and hospital have to continuously strive to manage clinical trainees by recognizing that infection prevention activities may be neglected as participants become less aware of COVID-19 due to continuous clinical practice. Participants complained of physical discomfort and fatigue as they had to wear a KF94 mask and goggles to prevent COVID-19, which is consistent with the nurses’ statement in the experience of nursing COVID-19 patients [32]. Therefore, it is necessary to prepare supplementary measures such as providing a space for rest during clinical practice and rest time. Communication in the medical environment is important in the medical delivery system and patient safety. As the use of KF94 masks increases to protect medical staff and patients during COVID-19, it has a negative effect on speech intelligibility [33], negatively affecting the communication and learning of nursing students in clinical practice with medical staff and patients. Therefore, efforts to develop communication and patient-centered nursing competency of nursing students [23] as a solution to identify the degree of difficulty in the therapeutic intervention due to limited non-verbal communication with patients by wearing masks. Participants were asked to take a passive practice attitude instead of an active practice attitude because of increased vigilance among members in the clinical setting due to concerns about COVID-19 infection [34]. Therefore, it can lead to a mismatch between learning goals and opportunities, impeding student competency achievement [35]. In order for students to actively accept clinical practice as an opportunity to develop themselves and not as a situation where they have to endure clinical practice [16], the university should strive to create a safe practice environment together with the hospital, instructors, and field leaders by providing a systematic clinical practice course.

In theme 5, “non-replaceable clinical field practice,” participants thought that nothing else could be a perfect substitute for clinical practice. Mutual interactions among members such as patients, caregivers, nurses, doctors, and student nurses continue in the clinical setting during processes of assessing the patients’ health status, establishing care plans, and performing and evaluating recovery plans for diseases and health promotion. Therefore, learning about effective communication at unpredictable clinical sites is difficult to achieve with only alternative clinical practice through video watching, case studies, and high-fidelity simulation [17]. Participants said that they obtained more up-to-date knowledge than that obtained from textbooks through clinical practice. They could comprehend the atmosphere of the clinical setting that they could not fully understand through instructors’ teachings or explanations from senior student nurses alone, such as attitudes as a nurse, understanding and responding to patients and caregivers, and establishing relationships with relevant departments [17]. They also gained pride as future nurses by recognizing the improvement of the positive image of professional nurses in clinical practice through patients, caregivers, and media during the COVID-19 pandemic [34]. The participants established their identity as nurses by trying to protect themselves and others through clinical interventions, which they were expected to do with anxiety amid the crisis of COVID-19. Clinical practice is an essential process for educating nurses so that nursing students can enter the clinical setting immediately after graduation. Therefore, nursing students also feel that no matter how many different contents they experience through online clinical practice, it is only an indirect experience. Such experience cannot completely replace the experience of seeing and learning directly from clinical practice [4].

Considering the results of this study, clinical field practice and alternative clinical practice in a pandemic situation have both positive and negative aspects for nursing students. We identified aspects of clinical practice that were not substituted by alternative clinical practices, such as in-school and online clinical practice. At the same time, the aspect of alternative clinical practice that compensates for the lack of clinical field practice was confirmed. Therefore, based on this, efforts should be made to establish a stable and systematic clinical practice system to improve the quality of clinical practice in the post-COVID-19 era.

One of the limitations of this study was that the data could not be generalized because the participants were recruited from three nursing schools only (out of 203 nursing schools in South Korea). We managed this limitation by diversifying the sample population in terms of clinical practice and geographical location (three different regions across the country). A follow-up study is required with unbiased, participation of nursing schools from more regions of the country, and more diversified training modalities.

## 5. Conclusions

Nursing students’ experiences of clinical practice and alternative clinical practice in the pandemic situation caused by COVID-19 can be expressed as “alienation during the process of clinical practice change” “regret caused by alternative clinical practice” “alternative clinical practice as a supplementary measure” “difficulties due to COVID-19” and “non-replaceable clinical field practice.” Based on this study, we would like to suggest the following proposals to prepare a strategy to improve the quality of clinical practice in the post-COVID-19 era.

First, it is necessary to provide opportunities for nursing students to participate as the main body of clinical practice in changing the clinical practice plan. Second, it is necessary to consider the continuous use of an alternative clinical practice to supplement clinical practice. Third, it is necessary to make full use of Edutech, such as Metaverse, to compensate for the shortcomings of alternative clinical practice, such as reduced concentration and lack of sense of presence. Fourth, it is necessary for all universities, faculty, field leaders, and hospitals to make an effort so that nursing students can receive systematic clinical practice education in a safe clinical setting, even during a crisis such as a pandemic.

## Figures and Tables

**Table 1 ijerph-19-13372-t001:** The semi-structured interview questionnaire.

Questions of the Interview
Tell me about the clinical practice you experienced in the COVID-19 pandemic.
What was the most difficult thing you experienced during clinical practice in the COVID-19 pandemic?
What was the best thing you experienced during clinical practice in the COVID-19 pandemic?
What do you think was the most important thing you experienced during clinical practice in the COVID-19 pandemic?
What do you think are the major changes in clinical practice caused by COVID-19 pandemic?

**Table 2 ijerph-19-13372-t002:** Theme and subthemes from student interview responses.

Themes	Subthemes
Alienation during the process of clinical practice change	Frustration aggravated by lack of communication
	Clinical practice took priority over student safety
Regret caused by alternative clinical practice	Repetition rather than critical thinking in virtual simulation
	Monotony due to fixed answers in case study
	Difficulty in concentration during watching the learning video
	Regret the lost opportunity to gain various experiences
Alternative clinical practice as a supplementary measure	Served as a preparation step before clinical field practice
	Feeling interested as the main body of clinical practice
	Satisfaction with a fair evaluation
	Less burden due to shortened clinical field practice period
Difficulties due to COVID-19	Anxiety about becoming a spreader
	Blurred awareness in the clinical setting
	Inconvenience caused by wearing protective equipment
	Exclusion from the clinical setting
Non-replaceable clinical field practice	Self-awareness of necessity
	Learning to build different relationships
	Establishment of identity as a future nurse

**Table 3 ijerph-19-13372-t003:** General characteristics of the participants.

Participant	Sex	Age (years)	COVID-19 Training Experience	Year 2020 Clinical Practice			Alternative Clinical Practice							
				CFP (Field)	ACT (Offline/School)	ACT (Online)	Off/S	On/S	PBL	CBL	CS	VM	Conf	ONSP
1	F	24	None	○	○	○	○	○	○	○	○	○	○	⨯
2	F	23	None	○	○	○	○	○	○	○	○	○	○	⨯
3	F	24	None	○	○	○	○	○	○	○	○	○	○	⨯
4	F	23	None	○	○	○	○	○	○	○	○	○	○	⨯
5	F	24	None	○	○	○	○	○	○	○	○	○	○	⨯
6	F	28	None	○	○	○	○	○	○	○	○	○	○	⨯
7	F	23	None	○	○	○	○	○	○	○	○	○	○	⨯
8	M	26	None	○	○	○	○	○	○	○	○	○	○	⨯
9	F	28	None	⨯	○	○	○	○	○	○	○	○	○	○
10	F	23	None	○	○	○	○	○	○	○	○	○	○	⨯
11	F	33	None	○	○	○	○	○	○	○	○	○	○	⨯
12	F	22	None	⨯	○	○	○	○	○	○	○	○	○	○
13	F	23	None	⨯	○	○	○	○	○	○	○	○	○	○
14	F	28	None	⨯	○	○	○	○	○	○	○	○	○	○

CFP: clinical field practice, ACT: alternative clinical training, Off/S: offline simulation, On/S: online simulation, PBL: problem-based learning, CBL: case-based learning, CS: case study, VM: video material, Conf.: conference, ONSP: online nursing skill program.

## Data Availability

The data presented in this study are available on request form the corresponding author.

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
