# Peer review of "Clinical Field and Alternative Clinical Practice Experience in a Pandemic Situation of Nursing Students Who Have Experienced Clinical Practice before COVID-19"

_ijerph, 2022, doi:10.3390/ijerph192013372_

Round 1
Reviewer 1 Report
First of all, congratulations on the work done. This is a subject that has clearly emerged from the COVID-19 pandemic and, as a nursing professor, is a theme that has led to a shift in our teaching paradigm.
Please find the following comments and recommendations as an opportunity to improve the manuscript.
Background
The background is well built, fluid, and logical. The topics are placed in an order that allows the reader to understand the rationale of the research question.
Methodology
The method is explicit, rigorous and adequate to the research problem that emerged. The study follows the COREQ reporting guidelines, which clearly helped in the article's development.
Results
In the theme "Regret about the tension that disappeared from clinical practice", the concept of tension does not appear to be what is specified in the interviews and the respective subthemes. Also, this theme is not very reader-friendly. I would suggest respecifying the theme.
"Aggravated difficulties due to COVID-19" would imply difficulties that already existed. Some of these difficulties are "new". Consider removing "aggravated".
Discussion
The discussion is well built, presenting an adequate confrontation between the findings and the literature.
Other findings
Could you please report if there were differences felt between the online and the face-to-face (offline) interviews?
The "Edutech" concept only emerges for the first time in the discussion. It would be beneficial to include the definition of this concept in the background.
Author Response
Point 1: In the theme "Regret about the tension that disappeared from clinical practice", the concept of tension does not appear to be what is specified in the interviews and the respective subthemes. Also, this theme is not very reader-friendly. I would suggest respecifying the theme.
Response 1: As you suggested, I respecified the theme.
Point 2: "Aggravated difficulties due to COVID-19" would imply difficulties that already existed. Some of these difficulties are "new". Consider removing "aggravated".
Response 2: As you suggested, I respecified the theme.
Point 3: Could you please report if there were differences felt between the online and the face-to-face (offline) interviews?
Response 3: Edited the data collection part for the additional clarification.
Point 4: The "Edutech" concept only emerges for the first time in the discussion. It would be beneficial to include the definition of this concept in the background.
Response 4: As you suggested, I edited the background.
Reviewer 2 Report
Conclusion: there is a number of limitations that need to be considered while interpreting this data. However, the authors are unaware of this. This lowers the value of this study. The conclusions are too important and the group is too small.
Author Response
Point 1: The conclusions are too important and the group is too small.
Response 1: This study selected the number of participants according to the qualitative research methodology. In this study, a total of 14 participants were recruited via purposive sampling. Data collection continued until theoretical saturation was reached. This study follows Consolidated Criteria for Reporting Qualitative Research (COREQ) reporting guidelines. As a qualitative study, this study tried to follow the evaluation criteria suggested by Guba and Lincoln(1981) in order to secure rigor. All procedures, including participant selection, were conducted in accordance with the qualitative research methodology.
Reviewer 3 Report
I thank the authors for the opportunity to review this interesting article. However, take into account the following recommendations and answer the questions raised:
• Cite the text properly, check line 67 of the text. It should be cited as follows [12-15]
• Please describe the theoretical framework used in the methodological design and justify the choice of phenomenology over other qualitative designs.
• In the choice of the sample, describe who is a junior and senior students, which course did the students belong to. Detail the exclusion criteria of the participants.
• Describe the academic context of the students: what is the duration of nursing studies in Korea? From what year do they practice? Are theoretical classes and clinical practices taught simultaneously? Where do students carry out practices?
• Detail how many interviews were carried out in person and how many by Zoom. What limitations did you find in the interviews that were conducted by Zoom? I understand that in the face-to-face interviews, the researcher and the students would wear a mask, to what extent the use of a mask influenced the data collection.
• The narratives of the participants should be aligned with the rest of the text, in this way it would improve the appearance of the manuscript.
All the best
Author Response
Point 1: Cite the text properly, check line 67 of the text. It should be cited as follows [12-15]
Response 1: Corrected
Point 2: Please describe the theoretical framework used in the methodological design and justify the choice of phenomenology over other qualitative designs.
Response 2: Edited the design and data collection part for the additional clarification.
Point 3: In the choice of the sample, describe who is a junior and senior students, which course did the students belong to. Detail the exclusion criteria of the participants.
Response 3: Edited the participant recruitment part for the additional clarification.
Point 4: Describe the academic context of the students: what is the duration of nursing studies in Korea? From what year do they practice? Are theoretical classes and clinical practices taught simultaneously? Where do students carry out practices?
Response 4: Edited the participant recruitment part for the additional clarification.
Point 5: Detail how many interviews were carried out in person and how many by Zoom. What limitations did you find in the interviews that were conducted by Zoom? I understand that in the face-to-face interviews, the researcher and the students would wear a mask, to what extent the use of a mask influenced the data collection.
Response 5: Edited the data collection part for the additional clarification.
Point 6: The narratives of the participants should be aligned with the rest of the text, in this way it would improve the appearance of the manuscript.
Response 6: Corrected
Round 2
Reviewer 2 Report
I made another detailed analysis. The research was conducted in accordance with COREQ. Research objective and results are consistent. However, in my opinion, the project has little scientific significance. But this is my personal opinion. Of course, qualitative research also has its value.
Reviewer 3 Report
After the changes introduced, the article has improved considerably and I believe that it can be published.